# The Cyclic Nitronate Route to Pharmaceutical Molecules: Synthesis of GSK’s Potent PDE4 Inhibitor as a Case Study

**DOI:** 10.3390/molecules25163613

**Published:** 2020-08-08

**Authors:** Evgeny V. Pospelov, Ivan S. Golovanov, Sema L. Ioffe, Alexey Yu. Sukhorukov

**Affiliations:** 1N.D. Zelinsky Institute of Organic Chemistry, Russian Academy of Sciences, 119991 Moscow, Russia; evpos00@mail.ru (E.V.P.); cell-25@yandex.ru (I.S.G.); iof@ioc.ac.ru (S.L.I.); 2Department of Chemistry, M.V. Lomonosov Moscow State University, 119991 Moscow, Russia; 3Department of Innovational Materials and Technologies Chemistry, Plekhanov Russian University of Economics, 117997 Moscow, Russia

**Keywords:** C–H functionalization, total synthesis, pyrrolidines, anchimeric assistance, epimerization, PDE4 inhibitors

## Abstract

An efficient asymmetric synthesis of GlaxoSmithKline’s potent PDE4 inhibitor was accomplished in eight steps from a catechol-derived nitroalkene. The key intermediate (3-acyloxymethyl-substituted 1,2-oxazine) was prepared in a straightforward manner by tandem acylation/(3,3)-sigmatropic rearrangement of the corresponding 1,2-oxazine-*N*-oxide. The latter was assembled by a (4 + 2)-cycloaddition between the suitably substituted nitroalkene and vinyl ether. Facile acetal epimerization at the C-6 position in 1,2-oxazine ring was observed in the course of reduction with NaBH_3_CN in AcOH. Density functional theory (DFT) calculations suggest that the epimerization may proceed through an unusual tricyclic oxazolo(1,2)oxazinium cation formed via double anchimeric assistance from a distant acyloxy group and the nitrogen atom of the 1,2-oxazine ring.

## 1. Introduction

Cyclic nitronates (1,2-oxazine-*N*-oxides **1** and isoxazoline-*N*-oxides **2**) are useful intermediates in the synthesis of complex nitrogen containing scaffolds due to their versatile reactivity as 1,3-dipoles and accessibility from nitroalkenes (Scheme 1a) [1,2,3,4,5,6,7,8,9,10]. Denmark’s group extensively exploited the inter- and intramolecular (3 + 2)-cycloaddition reactions with six-membered cyclic nitronates **1** to construct various bi- and polycyclic nitroso acetal frameworks **3**, which were then converted into fused pyrrolidine derivatives by an intramolecular reductive amination (Scheme 1b) [1,11]. Using this strategy, total syntheses of numerous pyrrolizidine and indolizidine alkaloids [12,13,14], as well as (5.5.5.5)- and (5.5.5.4)-azafenestanes [15,16], were accomplished.

Our group has a long-term interest in developing another approach towards the modification of cyclic nitronates, which utilizes C–H functionalization of the position next to the nitronate group (α-C-atom, Scheme 1c) [17]. Some time ago, we demonstrated that upon silylation, cyclic nitronates **1** and **2** are transformed into *N*-siloxyenamines **4**, in which the double bond is shifted to the exocyclic α-position [18]. Enamines **4** exhibit umpolung reactivity and react with nucleophiles in the presence of Lewis acids (LA) to give α-substituted cyclic oxime ethers **5** (1,2-oxazines or isoxazolines) via S_N_’ substitution of TMSO-group (Scheme 1c). Using this approach, nucleophilic halogenation [19], oxygenation [20,21,22], azidination [23] of cyclic nitronates were performed (route 1). Although we succeeded in using this methodology in the total synthesis of some pharmaceutical molecules, in the case of nitronates having acid-sensitive groups (e.g., acetals), it proved to be not very efficient (vide infra) [20,24,25].

Recently, we designed another strategy for the site-selective functionalization of cyclic nitronates **1** and **2** via acylation with acyl halides/Et_3_N (Scheme 1d) [26]. The initially formed *N*-acyloxyenamines **6** undergo a spontaneous (3,3)-rearrangement to give α-acyloxy-substituted cyclic oxime ethers **7** (route *2*). This tandem C–H oxygenation process could be performed under very mild conditions. At present, we are testing the scope and limitation of this method in the total synthesis of some model target molecules to compare its efficacy with our previous C–H functionalization methods. In this study, a potent GlaxoSmithKline’s phosphodiesterase 4 (PDE4) inhibitor **CMPO** [27,28,29], which was previously synthesized using route 1 [24,30], was chosen as a target molecule.

Scheme 2 depicts our previous synthetic route to **CMPO**. The fused pyrrolidine core was prepared by carbamylation of a *trans*-3-aryl-substituted prolinol **15**. The latter was accessed in our strategy by the reductive contraction of the 1,2-oxazine ring in the intermediate **14**, which was prepared by the stereoselective hydride reduction of the 5,6-dihydro-4*H*-1,2-oxazine **13** [24]. This intermediate was synthesized by the reduction of the ONO_2_-group in the corresponding nitroxy-substituted 1,2-oxazine **12**. Introduction of the nitroxy-group was accomplished by the LA-assisted functionalization of the methyl group in the *N*-oxide **10** using route 1 described above (see Scheme 1c). The required *N*-oxide **10** was assembled in a stereoselective manner by the (4 + 2)-cycloaddition of nitroalkene **8** with the vinyl ether **9** bearing Whitesell’s chiral auxiliary group ((+)-*trans*-2-phenyl-1-cyclohexanol ether).

Oxygenation of the methyl group in the *N*-oxide **10** proved to be challenging, and several methods were tested (Scheme 3). *N*-Siloxy,*N*-oxyenamine **16** was generated from the *N*-oxide **10** under mild conditions and then subjected to the LA-assisted nucleophilic addition of the bromide anion [24,25]. However, the desired 3-bromomethyl-1,2-oxazine **11**, which served as a precursor to nitrate **12**, was formed in moderate yields (best results are shown in Scheme 3a). Another issue was the epimerization of the sensitive C-6 acetal stereocenter leading to a mixture of 4,6-*trans*/4,6-*cis*-diastereomers **11** and **11′**, which had to be separated by column chromatography (Scheme 3a,b). Unfortunately, the epimer **11′** could not be used in the synthesis of **CMPO** as it produced the undesired 3,4-*cis*-stereoisomer upon the reduction of the C=N bond in 5,6-dihydro-4H-1,2-oxazine ring on the later stages of the synthesis (Scheme 3c) [25].

The reason for the epimerization may lie in the mechanism of the LA-promoted reaction of *N*-oxyenamines with nucleophiles, which involves heterolytic cleavage of the N–O bond (Scheme 3b). Experimental [20,22] and computational data [22] suggest that the S_N_’ substitution of the TMSO-group may proceed through an epimerizable *N*-vinyl,*N*-oxynitrenium cation **C1**. In our later study, we were able to optimize the epimerization ratio to 6: 1 by using Cr(NO_3_)_3_ as both a mild Lewis acid and the source of the nitrate anion [20]. However, the yield of nitroxy-derivative **12** was still not very high (ca. 40% from nitronate **10**, Scheme 3a). Thus, further optimization of the C–H functionalization stage was reasonable.

## 2. Results

We speculated that the pericyclic (3,3)-rearrangement of *N*-acyloxyenamine intermediate **I1** generated by the acylation of nitronate **10** may proceed without any epimerization of the C-6 stereogenic center. To test this idea, cyclic nitronate **10** was treated with pivaloyl chloride/Et_3_N (1.5/2.0 equiv.) under conditions previously optimized for model 1,2-oxazine-*N*-oxides (MeCN, −30 °C, 2 h) [26]. The desired pivalate **17** was formed in a 61% yield together with some amount of unreacted *N*-oxide **10**. After a short optimization of conditions, we found that the use of a bigger access of the PivCl/Et_3_N system (2.0/2.5 equiv.) and prolonged reaction time (18 h) at lower temperature resulted in an increase in the yield up to 76% (Scheme 4). Gratifyingly, no noticeable epimerization at the C-6 position was observed under these conditions.

We further investigated whether pivalate **17** could be used in the synthesis of **CMPO**. Hydrogenolysis of the pivalate group in **17** (to give alcohol **13**) prior the reduction is challenging since 5,6-dihydro-4H-1,2-oxazines are known to undergo fragmentation via a retro-[4+2]-cycloaddition process under the action of bases [31]. Therefore, pivalate **17** was subjected to the hydride reduction with NaBH_3_CN in acetic acid (Scheme 5). Surprisingly, the reaction produced two separable isomeric products **18** and **18′** in 3: 1 ratio (62% combined yield, 91% based on converted **17**). From the coupling constants in ^1^H-NMR spectra, it was deduced that both isomers had *trans*-arrangement of the substituents at the C-3 and C-4 atoms, while the configuration of the C-6 stereocenter was different. Thus, the C-6 acetal moiety underwent epimerization in the course of the reduction (see Discussion section). The amount of 4,6-*cis*-isomer **18′** increased with time, demonstrating that the isomerization took place in the reduced product **18** and not in the starting compound **17**. This is also confirmed by the fact that stereisomers **18** and **18′** had same configuration of the newly formed C-3 stereocenter. If the epimerization preceded the reduction, the C-6 epimerized 5,6-dihydro-4H-1,2-oxazine **17′** would also give the reduced product with the 3,4-*cis*-disposition of substituents (see Scheme 3c).

Isomers **18** and **18′** could be almost equally used in the reductive contraction of the 1,2-oxazine ring [32], since both produced the same amino aldehyde intermediate **I3** upon cleavage of the N–O bond followed by fragmentation of the hemiacetal **I2** (Scheme 6). Subsequent intramolecular reductive amination in the intermediate **I3** and protection with the Boc-group afforded the desired prolinol ester **19**. Hence, the separation of epimers **18** and **18′** was not required and a mixture could be converted into the product **19** in a 61% yield. The chiral auxiliary alcohol (*trans*-2-phenylcyclohexanol) was recovered at this stage in 77% yield.

On the next step, careful saponification of the pivalate moiety in pyrrolidine **19** with KOH in aqueous methanol gave Boc-prolinol **20** in 84% yield (Scheme 6). It is noteworthy that hydrolysis of the Boc-group was also observed to some extent under these conditions. For this reason, the reaction mixture was treated with Boc_2_O after neutralization to convert the unprotected prolinol into the *N*-Boc-derivative **20**, which was isolated by column chromatography.

Finally, deprotection of the *N*-Boc moiety with TFA and treatment of the resulting prolinol trifluoroacetate with Im_2_CO/Et_3_N afforded the desired PDE4 inhibitor **CMPO** (Scheme 6). Thus, the asymmetric synthesis of PDE4 inhibitor **CMPO** was completed in seven steps from a known nitroalkene **8** in 8% overall yield. Chiral HPLC analysis revealed the enantiomeric purity of the product > 97% *ee*. The racemic sample of **CMPO** for HPLC analysis was prepared according to the same synthetic sequence starting from the racemic *trans*-2-phenylcyclohexanol.

## 3. Discussion

Epimerization of the acetal moiety in the course of the hydride reduction of 5,6-dihydro-4-H-1,2-oxazine **17** is of special note. In the previously reported hydride reduction of 1,2-oxazine **13** possessing a free hydroxymethyl group, no epimerization at the C-6 atom was observed (cf. data in Scheme 2; Scheme 5) [24]. We hypothesized that such a difference in the behavior of 3-hydroxymethyl- and 3-acyloxymethyl-substituted 1,2-oxazines **13** and **17** may be attributed to an anchimeric assistance from the carbonyl group, which stabilizes the intermediate cation **C2** [33] by forming a bridged system with an eight-membered ring (cation **C3**). In carbohydrates, a similar anchimeric assistance of the acyloxy group from the distant 1,4-position has been proposed, yet it was not confirmed unambiguously by experimental data [34,35,36,37]. In our case, density functional theory (DFT) calculations at the MN15/Def2TZVP level of theory (see Appendix A for details) revealed that the bridged cation **C3** is much less stable compared to the initial monocyclic cation **C2**. Interestingly, the formation of a third ring between the nitrogen atom and the acyloxy group may lead to a tricyclic cation **C4**, which is predicted to be much more stable than the mono- or bicyclic structures **C2** and **C3** (Scheme 7). The formation of such a stable tricyclic cation as an intermediate or a resting state may account for the observed facile epimerization of pivalate **18**. The higher thermodynamic stability of the 4,6-*cis*-isomer **18′** over the 4,6-*trans*-isomer **18**, as shown by DFT calculations, is likely to be the driving force for the epimerization at the C-6 atom.

Another remarkable observation was that 1,2-oxazine **18** (as well as its precursor **17**) did not undergo epimerization in acetic acid (rt, 2 h). Isomerization to the *cis*-isomer **18′** was observed only in the presence of NaBH_3_CN. Hence, the fragmentation of the acetal moiety is most likely promoted by some Lewis acidic boron species generated from NaBH_3_CN in acidic medium. Indeed, slow epimerization of 1,2-oxazine **18** was observed upon treatment of **18** with B(OBu)_3_ or BF_3_·Et_2_O in acetic acid.

It is noteworthy that an anchimeric-assisted epimerization in 1,2-oxazine series has not been reported previously. Moreover, to our knowledge, this is the first reported example of a remote neighboring group participation from the 1,4-position in a six-membered ring confirmed by DFT calculations [38,39,40]. The formation of an eight-membered ring in this case may be driven by an unusual secondary anchimeric interaction involving the nitrogen atom of the 1,2-oxazine ring leading to an unusual tricyclic oxazolo(1,2)oxazinium cation **C4**.

## 4. Materials and Methods

All reactions were carried out in oven-dried (150 °C) glassware. NMR spectra (Bruker AM 300 spectrometer, Karlsruhe, Germany) were recorded at room temperature (if not stated otherwise) with residual solvents peaks as an internal standard. Peak multiplicities are indicated by s (singlet), d (doublet), t (triplet), dd (doublet of doublets), q (quartet), quint (quintet), ddd (doublet of doublets of doublets), tt (triplet of triplets), tdd (triplets of doublets of doublets), m (multiplet), br (broad). The numeration of atoms used in the assignment of NMR spectra is given in Figure 1.

HRMS were measured on the electrospray ionization (ESI) (Bruker MicroTOF, Karlsruhe, Germany) instrument with a time-of-flight (TOF) detector. Concentrations c in optical rotation angles are given in g/100 mL. [α]_D_ values are given in 10^−1^ deg cm^2^ g^−1^. Column chromatography was performed using Kieselgel (Merck, Germany) 40–60 μm 60A. Analytical thin-layer chromatography was performed on silica gel plates with QF 254. Visualization was accomplished with UV light and solution of anisaldehyde/H_2_SO_4_ in ethanol. Chiral HPLC analysis was performed on a chromatograph with a UV-VIS photodiode array detector (Shimadzu LC-20, Shimadzu, Japan). All reagents were commercial grade and used as received. Et_3_N, dichloromethane (DCM), and MeCN were distilled over CaH_2_ prior to the experiments; MeOH, hexane and ethyl acetate were distilled without drying agents.

*(+)-(4S,6S)-4-(3-(Cyclopentyloxy)-4-methoxyphenyl)-3-methyl-6-(((1S,2R)-2-phenylcyclohexyl)oxy)-5,6-dihydro-4H-1,2-oxazine 2-oxide (+)-***10** and the corresponding racemate rac-**10** were synthesized according to the previously reported procedure from nitroalkene **8** and vinyl ether **9 [25]**. In most experiments, very similar yields in enantiomeric and racemic series were obtained. In Schemes and Procedures, the best yields among two series are given if not otherwise stated.

Oxygenation of nitronate **10**. Enantipure or racemic nitronate **10** (347 mg, 0.72 mmol) was dissolved in dry acetonitrile (1.7 mL) in a Schlenk tube under argon atmosphere, and then Et_3_N (251 μL, 1.8 mmol) was added. The solution was cooled to −40 °C and pivaloyl chloride (174 μL, 1.41 mmol) was added. The reaction mixture was stirred at ca. −40 °C for 2 h and then kept in a freezer (ca. −25 °C) overnight. The mixture was diluted with EtOAc (5 mL) and transferred into a separating funnel containing EtOAc (20 mL) and 0.25 M aq. NaHSO_4_ solution (20 mL). The aqueous layer was extracted with EtOAc (20 mL), the combined organic layers were washed with water (30 mL) and brine (30 mL), dried over anhydrous Na_2_SO_4_ and concentrated under reduced pressure. The residue was subjected to a column chromatography on silica gel (Hexane/EtOAc = 10/1) to give 311 mg (76%) of enantiopure or racemic pivalate **17**.

*(1S,2R,4S,6S)*- and *(1S*,2R*,4S*,6S*)-(4-(3-(cyclopentyloxy)-4-methoxyphenyl)-6-((2-phenylcyclohexyl)oxy)-5,6-dihydro-4H-1,2-oxazin-3-yl)methyl pivalate* (**17**). R_f_ = 0.44 (hexane/EtOAc = 3/1). ^1^H-NMR (300 MHz, COSY, HSQC, CDCl_3_) δ 7.41–7.13 (m, 5H, H14, H15, H16), 6.75 (d, *J* = 8.2 Hz, 1H, H23), 6.57 (dd, *J* = 8.2, 2.1 Hz, 1H, H22), 6.53 (d, *J* = 2.2 Hz, 1H, H26), 5.36 (dd, *J* = 2.7, 2.5 Hz, 1H, H_eq_6), 4.71 (m, 1H, H28), 4.08 (d, *J* = 13.5 Hz, 1H, H17′), 4.02 (ddd, *J* = 10.3, 10.2, 4.0 Hz, 1H, H_ax_7), 3.98 (d, *J* = 13.5 Hz, 1H, H17′’), 3.80 (s, 3H, H27), 2.92 (dd, *J* = 11.9, 7.7 Hz, 1H, H_ax_4), 2.61 (ddd, *J* = 11.1, 10.3, 3.6 Hz, 1H, H_ax_8), 2.37 (m, 1H, H12), 2.01 (ddd, *J* = 13.0, 7.7, 2.7 Hz, 1H, H_eq_5), 1.98–1.73 (m, 10H, H_ax_5, H9′, H10, H29, H30′), 1.70–1.52 (m, 3H, H9”, H30′’), 1.50–1.28 (m, 3H, H11, H_eq_12), 1.10 (s, 9H, H20). ^13^C-NMR (75 MHz, HSQC, CDCl_3_) δ 177.4 (C18), 155.3 (C3), 149.3 and 147.9 (C24 and C25), 144.4 (C13), 131.1 (C21), 128.0 and 127.8 (2 C14 and 2 C15), 125.9 (C16), 120.7 (C22), 114.9 (C26), 112.2 (C23), 91.0 (C6), 80.4 (C28), 76.1 (C7), 63.6 (C17), 56.1 (C27), 50.7 (C8), 38.6 (C19), 34.3 (C9), 33.8 (C4), 32.8 and 32.7 (C29 and C29′), 32.5 (C5), 30.6 (C12), 27.1 (3 C20), 26.1 (C11), 24.7 (C10), 24.0 (C30 and C30′). HRMS (ESI): *m/z* calcd. for [C_34_H_46_NO_6_]^+^ 564.3320, found 564.3316 [M + H]^+^.

(+)-(1*S*,2*R*,4*S*,6*S*)-**17**. Colorless oil. [α]_D_ = +188.6 (c = 0.09, EtOAc, 20 °C). *rac*-**17**. Colorless oil.

Hydride reduction of 5,6-dihydro-4H-1,2-oxazine **17**. *Procedure 1:* Enantiopure or racemic pivalate **17** (80 mg, 0.14 mmol) was dissolved in acetic acid (0.8 mL) and sodium cyanoborohydride (120 mg, 1.9 mmol) was added to the solution upon intensive stirring. The reaction mixture was stirred under argon for 30 min at rt, then diluted with EtOAc (3 mL) and transferred into a separating funnel containing EtOAc (20 mL) and a sat. aq. NaHCO_3_ solution (20 mL). The aqueous layer was extracted with EtOAc (20 mL). The combined organic layers were washed with sat. aq. NaHCO_3_ solution (20 mL) and brine (40 mL), then dried over Na_2_SO_4_ and concentrated under reduced pressure. The residue was subjected to a column chromatography on silica gel (Hexane/EtOAc = 10/1→5/1→3/1) to yield 49 mg (62%) of a mixture of 1,2-oxazines **18** and **18′** (d.r. 3: 1). Also, 26 mg (33%) of unreacted pivalate **17** was isolated from the column chromatography.

*Procedure 2:* Enantiopure or racemic pivalate **17** (190 mg, 0.34 mmol) was dissolved in acetic acid (1.8 mL) and sodium cyanoborohydride (250 mg, 3.97 mmol) was added to the solution upon intensive stirring. The reaction mixture was stirred under argon at rt for 1.7 h, then diluted with EtOAc (3 mL) and transferred into a separating funnel containing EtOAc (20 mL) and a sat. aq. NaHCO_3_ solution (20 mL). The aqueous layer was extracted with EtOAc (20 mL). The combined organic layers were washed with sat. aq. NaHCO_3_ solution (20 mL) and brine (40 mL), then dried over Na_2_SO_4_ and concentrated under reduced pressure. The residue was subjected to a column chromatography on silica gel (Hexane/EtOAc = 10/1→5/1→3/1) to yield 48 mg (25%) of the fast moving isomer **18** and 69 mg (36%) of the slow moving isomer **18′**.

*((3R,4S,6S)- and (3R*,4S*,6S*)-4-(3-(Cyclopentyloxy)-4-methoxyphenyl)-6-(((1S,2R)-2-phenylcyclohexyl)oxy)-1,2-oxazinan-3-yl)methyl pivalate* (**18**). Colorless oil (both enantiopure and racemic). R_f_ = 0.37 (hexane/EtOAc = 3/1). ^1^H-NMR (300 MHz, COSY, HSQC, CDCl_3_) δ 7.55–7.32 (m, 4H, H14, H15), 7.29–7.21 (m, 1H, H16), 6.76 (d, *J* = 8.0 Hz, 1H, H23), 6.61 (dd, *J* = 8.0, 2.1 Hz, 1H, H22), 6.59 (br s, 1H, H26), 4.99 (d, *J* = 3.0 Hz, 1H, H6), 4.79–4.68 (m, 1H, H28), 4.03 (br d, *J* = 10.7 Hz, 1H, H_ax_2), 3.80 (s, 3H, H27), 3.78 (ddd, *J* = 10.0, 10.0, 3.9 Hz, 1H, H_ax_7), 3.51 (dd, *J* = 11.8, 2.8 Hz, 1H, H17′), 3.22 (br ddd, *J* = 10.7, 9.8, 8.5 Hz, 1H, H_ax_3), 3.08 (dd, *J* = 11.8, 8.5 Hz, 1H, H17′’), 2.68 (ddd, *J* = 12.3, 10.3, 3.7 Hz, 1H, H_ax_8), 2.32 (ddd, *J* = 11.9, 11.9, 4.5 Hz, 1H, H_ax_4), 2.21 (br d, *J* = 10.6 Hz, 1H, H_ax_12), 2.05–1.75 (m, 11H, H5, H9′, H10, H29, H30′), 1.70–1.54 (m, 3H, H9”, H30′’), 1.48–1.25 (m, 3H, H11, H_eq_12), 1.12 (s, 9H, H20). ^13^C-NMR (75 MHz, HSQC, HMBC, DEPT135, CDCl_3_) δ 178.1 (C18), 149.1 and 147.8 (C24 and C25), 144.5 (C13), 133.9 (C21), 128.8 and 128.0 (2 C14 and 2 C15), 126.9 (C16), 119.5 (C22), 114.2 (C26), 112.3 (C23), 93.5 (C6), 80.5 (C28), 77.7 (C7), 63.3 (C17), 60.1 (C3), 56.1 (C27), 50.7 (C8), 38.6 (C19), 37.2 (C4), 37.0 (C5), 33.3 (C9), 32.7 and 32.8 (C29 and C29′), 31.3 (C12), 27.1 (3 C20), 26.0 (C11), 24.8 (C10), 24.0 (C30 and C30′). HRMS (ESI): *m/z* calcd. for [C_34_H_48_NO_6_]^+^ 566.3476, found 566.3474 [M + H]^+^.

*((3R,4S,6R)- and (3R*,4S*,6R*)-4-(3-(Cyclopentyloxy)-4-methoxyphenyl)-6-(((1S,2R)-2-phenylcyclohexyl)oxy)-1,2-oxazinan-3-yl)methyl pivalate* (**18′**). Characterized in mixture with **18** (d.r. **18**/**18′** = 1: 4). Colorless oil (both enantiopure and racemic). R_f_ = 0.30 (hexane/EtOAc = 3/1). ^1^H-NMR (300 MHz, COSY, HSQC, CDCl_3_) δ 7.55–7.32 (m, 5H, H14, H15, H16), 6.79 (d, *J* = 8.2 Hz, 1H, H23), 6.60 (dd, *J* = 8.2, 2.2 Hz, 1H, H22), 6.55 (d, *J* = 2.2 Hz, 1H, H26), 5.43 (d, *J* = 10.4 Hz, 1H, H2), 5.28 (d, *J* = 2.9 Hz, 1H, H6), 4.73 (m, 1H, H28), 3.93 (ddd, *J* = 10.3, 10.2, 4.0 Hz, 1H, H7), 3.81 (s, 3H, H27), 3.59 (d, *J* = 12.9 Hz, 1H, H17), 3.48 (ddd, *J* = 11.2, 10.4, 5.7 Hz, 1H, H_ax_3), 3.38 (dd, *J* = 12.9, 5.7 Hz, 1H, H17), 2.71 (ddd, *J* = 10.6, 10.3, 3.3 Hz, 1H, H_ax_8), 2.63 (ddd, *J* = 12.5, 11.2, 4.0 Hz, 1H, H_ax_4), 2.24 (br d, *J* = 11.9 Hz, 1H, H_ax_12), 2.06 (td, *J* = 13.3, 12.5, 2.9 Hz, 1H, H_ax_5), 1.99–1.76 (m, 10H, H_eq_5, H9′, H10, H29, H30′), 1.72–1.53 (m, 3H, H9”, H30′’), 1.53–1.25 (m, 3H, H11, H_eq_12), 1.10 (s, 9H, H20). ^13^C-NMR (75 MHz, HSQC, HMBC, DEPT135, CDCl_3_) δ 177.4 (C18), 149.9 and 148.2 (C24 and C25), 143.7 (C13), 130.6 (C21), 129.6 (C16), 127.7 and 127.5 (2 C14 and 2 C15), 120.0 (C22), 113.7 (C26), 112.4 (C23), 95.2 (C6), 80.7 (C28), 78.5 (C7), 65.8 (C3), 61.3 (C17), 56.0 (C27), 50.5 (C8), 38.5 (C19), 36.7 (C4), 36.4 (C5), 33.1 (C9), 32.8 and 32.7 (C29 and C29′), 30.7 (C12), 27.0 (3 C20), 25.8 (C11), 24.4 (C10), 24.0 (C30 and C30′). HRMS (ESI): *m/z* calcd. for [C_34_H_48_NO_6_]^+^ 566.3476, found 566.3476 [M + H]^+^.

Hydrogenation of 1,2-oxazines **18** and **18′**. A glass vial was charged with a solution of enantiopure 1,2-oxazine **18** (48 mg, 0.086 mmol) and Boc_2_O (58 mg, 0.264 mmol) in methanol (0.5 mL). A suspension of Raney nickel (ca. 100 mg, prepared from 50% slurry in water) in methanol (ca. 0.5 mL) was added, and the vial was placed in a steel autoclave, which was then flushed and filled with hydrogen to a pressure of ca. 40 bar and heated to 50 °C. The hydrogenation was conducted for 2 h with intensive stirring. Then, the autoclave was cooled to rt, slowly depressurized, and the catalyst was removed using a magnet and washed with methanol. The solution was concentrated to dryness under reduced pressure. The residue was subjected to a column chromatography on silica gel (hexane/EtOAc = 20/1→10/1) to yield 31 mg (74%) of *N*-Boc pyrrolidine **19**. The column was then washed with hexane/EtOAc = 5/1 to recover (+)-*trans*-2-phenylcyclohexanol (12 mg, 77%).

Application of the same procedure for the reduction of enantiopure 1,2-oxazine **18′** (69 mg, 0.13 mmol) afforded 36 mg (60%) of *N*-Boc pyrrolidine **19** and 15 mg (66%) of (+)-*trans*-2-phenylcyclohexanol.

Application of the same procedure for the reduction of a mixture of racemic 1,2-oxazines *rac*-**18** and *rac*-**18′** (90 mg, 0.159 mmol, d.r. 1: 1.4) afforded 46 mg (61%) of racemic *N*-Boc pyrrolidine *rac*-**19**.

*Tert-butyl (2R,3S)- and (2R*,3S*)-3-(3-(cyclopentyloxy)-4-methoxyphenyl)-2-((pivaloyloxy)methyl)pyrrolidine-1-carboxylate* (**19**). *R_f_* = 0.30 (hexane/EtOAc = 3/1). ^1^H-NMR (300 MHz, 320K, COSY, HSQC, CDCl_3_) δ 6.82 (d, *J* = 8.7 Hz, 1H, H15), 6.75–6.70 (m, 2H, H14 and H21), 4.76 (m, 1H, H18), 4.41–4.26 (m, 1H, H9′), 4.20 (br d, *J* = 9.9 Hz, 1H, H9′’), 4.11–3.88 (br m, 1H, H2), 3.84 (s, 3H, H17), 3.77–3.62 (br m, 1H, H5′), 3.46–3.31 (m, 1H, H5′’), 3.29–3.10 (br m, 1H, H3), 2.35–2.18 (m, 1H, H4′), 2.01–1.76 (m, 7H, H4′’, H19, H20′), 1.72–1.54 (m, 2H, H20′’), 1.49 (s, 9H, H8), 1.21 (s, 9H, H12). ^13^C-NMR (75 MHz, HSQC, DEPT135, CDCl_3_) δ 178.2 (C10), 154.2 (C6), 148.9 and 147.9 (C16 and C22), 135.1 (C13), 119.2 (C14), 114.0 (C21), 112.3 (C15), 80.5 (C18), 80.0 (C7), 63.5 (br, C9), 63.1 (br, C2), 56.2 (C17), 47.2 (C3), 46.3 (C5), 38.8 (C11), 32.8 (C19 and C19′), 31.8 (C4), 28.5 (3 C8), 27.2 (3 C12), 24.0 (C20 and C20′) (signals are broadened due to the presence of *N*-Boc rotamers). HRMS (ESI): *m*/*z* calcd. for [C_27_H_42_NO_6_]^+^ 476.3007, found 476.3006 [M + H]^+^.

(−)-(2*R*,3*S*)-**19**. Colorless oil, [α]_D_ = −19.0 (*c* = 1, EtOAc, 24 °C). *rac*-**19**. Colorless oil.

Saponification of pivalate **19**. Enantiopure or racemic pivalate **19** (67 mg, 0.14 mmol) was dissolved in MeOH (2.8 mL) and a solution of KOH (237 mg, 4.2 mmol) in H_2_O (1.4 mL) was added. The mixture was stirred at room temperature for 24 h. Then, acetic acid (0.4 mL) was added and the reaction mixture was stirred for 5 min. The resulting solution was concentrated in vacuum. To the residue, MeOH (2 mL) and Boc_2_O (0.065 g, 0.28 mmol) were added and the resulting solution was stirred for 1 h. Then, volatiles were removed in vacuum and the residue was subjected to a column chromatography on silica gel (hexane/EtOAc = 5/1→3/1→1/1) to yield 46 mg (84%) of prolinol **20**.

*Tert-butyl (2R,3S)- and (2R*,3S*)-3-(3-(cyclopentyloxy)-4-methoxyphenyl)-2-(hydroxymethyl)pyrrolidine-1-carboxylate* (**20**). *R_f_* = 0.21 (hexane/EtOAc = 1/1). ^1^H-NMR (300 MHz, COSY, HSQC, CDCl_3_) δ 6.81 (d, *J* = 7.6 Hz, 1H, H13), 6.76 (d, *J* = 7.6 Hz, 1H, H12), 6.74 (s, 1H, H16), 5.13–4.88 (br, 1H, H10), 4.77 (m, 1H, H18), 3.97–3.85 (br m, 1H, H2), 3.83 (s, 3H, H17), 3.80–3.66 (br m, 2H, H9′, H5′), 3.63 (dd, *J* = 11.5, 6.8 Hz, 1H, H9′’), 3.44–3.27 (br m, 1H, H5′’), 2.89–2.79 (br m, 1H, H3), 2.22–2.07 (br m, 1H, H4′), 2.02–1.75 (m, 7H, H20′, H19, H4′’), 1.67–1.57 (m, 2H, H20′’), 1.51 (s, 9H, H8). ^13^C-NMR (75 MHz, HSQC, DEPT135, CDCl_3_) δ 156.8 (C6), 149.2 and 147.9 (C14 and C15), 133.3 (C11), 119.7 (C12), 114.5 (C16), 112.3 (C13), 80.5 (C18), 80.4 (C7), 67.1 (C2), 66.1 (C9), 56.2 (C17), 47.4 (C3), 47.0 (C5), 32.9 and 32.8 (C-4, C19 and C19′), 28.5 (3 C8), 24.0 (C20 and C20′). HRMS (ESI): *m*/*z* calcd. for [C_22_H_33_NO_5_Na]^+^ 414.2251, found 414.2247 [M + Na]^+^.

(−)-(2*R*,3*S*)-**20**. Colorless oil, [α]_D_ = −9.3 (*c* = 1, EtOAc, 24 °C). *rac*-**20**. Colorless oil.

Synthesis of (−)*-* and *rac-***CMPO**. To a stirred solution of enantiopure or racemic prolinol **20** (45 mg, 0.115 mmol) in DCM (0.9 mL) was added CF_3_COOH (0.18 mL, 2.4 mmol) at 0–5 °C. The cooling bath was removed, and the solution was stirred for 1 h. Then, volatiles were removed under reduced pressure and the residue was dried until constant weight. The resulting trifluoroacetate was dissolved in DCM (0.85 mL), and Et_3_N (0.08 mL, 0.58 mmol) and 1,1′-carbonyldiimidazole (47 mg, 0.29 mmol) were added at rt. The solution was stirred for 18 h at rt, and then concentrated under reduced pressure. The product was isolated by column chromatography on silica gel (hexane/EtOAc = 3/1) followed by recrystallization from hexane\diethyl ether (ca. 1: 1). Yield: 18 mg (49%). ^1^H NMR spectra were in agreement with previously published data [24].

*(−)-(7S,7aR)*-**CMPO**. White solid. Mp = 134–137 °C (lit.[24] 137–139 °C). HPLC analysis: *ee* > 97% (RT 9.8 min; column CHIRALPAK IA-3 (15 cm); solvent Hexane/*i*-PrOH = 90:10; temperature 40 °C; flow rate 1 mL/min). [α]_D_ = −63.0 (*c* = 0.5, EtOAc, 25 °C). lit.[24] [α]_D_ = −69.1 (*c* = 0.83, MeOH, 26 °C).

*rac-***CMPO**. White solid. Mp = 103–104 °C (lit.[30] 99–101 °C). HPLC analysis: (+)-(7*R*,7a*S*)-**CMPO** (RT 8.8 min) and (−)-(7*S*,7a*R*)-**CMPO** (RT 9.8 min); column CHIRALPAK IA-3 (15 cm); solvent Hexane/*i*-PrOH = 90:10; temperature 40 °C; flow rate 1 mL/min.

## 5. Conclusions

In conclusion, we were able to solve the problem of site-selective C–H oxygenation of the cyclic nitronate intermediate in the asymmetric synthesis of a potent PDE4 inhibitor **CMPO** by using tandem acylation/(3,3)-sigmatropic rearrangement. In comparison with the previous synthesis, this method afforded the required 3-oxymethyl-substituted 1,2-oxazine intermediate in a much higher yield (76% vs. 27%). This key intermediate could be readily converted into the target (−)-**CMPO** by the reductive contraction of the 1,2-oxazine ring followed by deprotection and carbamylation with Im_2_CO. A rapid epimerization of the C-6 acetal moiety was observed upon the reduction of the 5,6-dihydro-4H-1,2-oxazine ring with NaBH_3_CN in acetic acid. DFT calculations suggest that the epimerization is favored by an unprecedented double anchimeric assistance from a remote acyloxy group and the nitrogen atom of the 1,2-oxazine ring.

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
