# Peer review of "The Cyclic Nitronate Route to Pharmaceutical Molecules: Synthesis of GSK’s Potent PDE4 Inhibitor as a Case Study"

_molecules, 2020, doi:10.3390/molecules25163613_

Round 1

Reviewer 1 Report

This report by Sukhorukov and coworkers presents a revised route to (-)-CMPO through nitrone rearrangement, and presents DFT work to help explain the selectivity. I think the chemistry is nicely presented, and although I'm not convinced of the novelty and significance, I do think overall the study is well done and is appropriate for Molecules. The SI shows reasonably clear and clean spectra, though the authors should check that they have consistently labeled peaks (i.e. S5 - some peaks are not integrated such as the OMe group). My only concern is that 16 of 40 citations seem to be self-citations. I am not so familiar with this area to know whether or not there is significant relevant work that has been ignored in favor of self-citation, but I would definitely suggest that this level of self-citation is very high, and may need to be changed or justified to the editor.

Author Response

We would like to thank you for a very helpful revision of our manuscript. The answers to your comments are given below:

1) Reviewer’s comment: The SI shows reasonably clear and clean spectra, though the authors should check that they have consistently labeled peaks (i.e. S5 - some peaks are not integrated such as the OMe group).

Answer: Thank you. We checked SI and added missed integration in NMR spectra on pages S5 and S28.

2) Reviewer’s comment: My only concern is that 16 of 40 citations seem to be self-citations.

Answer: Thank you for this suggestion. Since this manuscript is a part of a large project on the application of the nitronate CH-functionalization in the target-oriented synthesis, we had to cite our previous works related to the project. For this reason the number of self-citations turned out to be so high. Following your suggestion, in the revised manuscript we reduced the number of self-citations to 10 (six of self-citations were changed to other relevant references). Numeration of references was changed accordingly.

Reviewer 2 Report

Generally, the manuscript is well written with sufficient background and clear explanations. It demonstrates an interesting and efficient asymmetric synthesis of a complex and pharmaceutically relevant compound with the recently developed acylation/[3,3]‐sigmatropic rearrangement as the key transformation in the proposed synthetic sequence.

In my opinion, the section from Page 2, line 58 to Page 4, line 96 (including Schemes 2 and 3) describes previously published material, so the authors should move this to the "Introduction" in the revised version. The actual results relevant to the manuscript start on Page 4, line 98 with the sentence: "We speculated, that the pericyclic [3,3]‐rearrangement of N‐acyloxyenamine intermediate...."

Can the authors comment on the possibility of mild hydrolysis of 17 to 13 in order to remove pivaloyl group without cleaving the acetal at C-6, then followed by the reductive ring contraction of 1,2-oxazine ring in 13 to 14 where C-6 epimerization was not an issue? Although the C-6 epimerization is not problematic for the rest of the sequence (since both 18 and 18' give the same product with the correct stereochemistry at C-3), perhaps this approach would avoid using Boc-protection and extra Boc2O to compensate for the partial loss of N-Boc group during Piv hydrolysis (19 to 20), and reduce the total number of steps to 7.

Why did the authors choose to use MN15/Def2TZVP level of theory and SMD model for solvation calculations? How do these results compare to IEFPCM or CPCM methods? Also, it is not clear what the Lewis acid was (if any; although shown in Scheme 7?).

Author Response

We would like to thank you for a very helpful revision of our manuscript. The answers to your comments are given below:

1) Reviewer’s comment: In my opinion, the section from Page 2, line 58 to Page 4, line 96 (including Schemes 2 and 3) describes previously published material, so the authors should move this to the "Introduction" in the revised version. The actual results relevant to the manuscript start on Page 4, line 98 with the sentence: "We speculated, that the pericyclic [3,3]‐rearrangement of N‐acyloxyenamine intermediate...."

Answer: Thank you for this suggestion. The discussion of previous results (Schemes 2, 3, and the describing text) were moved to the Introduction section of the manuscript.

2) Reviewer’s comment: Can the authors comment on the possibility of mild hydrolysis of 17 to 13 in order to remove pivaloyl group without cleaving the acetal at C-6, then followed by the reductive ring contraction of 1,2-oxazine ring in 13 to 14 where C-6 epimerization was not an issue? Although the C-6 epimerization is not problematic for the rest of the sequence (since both 18 and 18' give the same product with the correct stereochemistry at C-3), perhaps this approach would avoid using Boc-protection and extra Boc2O to compensate for the partial loss of N-Boc group during Piv hydrolysis (19 to 20), and reduce the total number of steps to 7.

Answer: Thank you for an interesting suggestion. The removal of the pivalate group requires strong alkaline conditions (e.g. LiOH). Unfortunately, 5,6-dihydro-4H-1,2-oxazines (such as 17 and 13) are sensitive to bases, which deprotonate the enolizable C-4 position (please, see Scheme in the attachment). This provokes subsequent fragmentation of the 1,2-oxazine ring via a retro-[4+2]-cycloaddition process (Bull. Chem. Soc. Jpn., 1988, 61, 461; Synthesis, 2007, 97). Nevertheless, during optimization studies we tried to perform saponification of the pivalate group on a model 5,6-dihydro-4H-1,2-oxazine (similar to 17, but without a chiral auxiliary group). However, a complex mixture of products was obtained. A comment on this was added in the manuscript text (as note 31).

3) Reviewer’s comment: Why did the authors choose to use MN15/Def2TZVP level of theory and SMD model for solvation calculations? How do these results compare to IEFPCM or CPCM methods? Also, it is not clear what the Lewis acid was (if any; although shown in Scheme 7?).

Answer: MN15 functional performs well in comparison with other popular functionals (B3LYP, M06 for example) for organic molecules geometries (SE47 database) and activation barriers (BH76 database). Authors of this new functional suppose its broad accuracy for various properties (Chem. Sci., 2016, 7, 5032-5051). Def2TZVP basis set is rather popular and it can be used to obtain results that are not too far from the DFT basis set limit (Phys. Chem. Chem. Phys., 2005, 7, 3297-3305). SMD model is a modification of IEFPCM model (J. Phys. Chem. B 2009, 113, 18, 6378–6396). Also it was developed and tested to be used in conjunction with Minnesota functionals family (MN15 functional is a part of this family). Lewis acid was not included in the calculations (optimization of geometry and energy calculations were conducted for cationic structures C2 – C4 without the counterion containing a Lewis acid). The corresponding statement was added in the caption of Scheme 7.

Reviewer 3 Report

The authors  described the methodorogy of pyrrolidine synthesis via the cyclic nitronate route.
The authors have aovercome the issue of previous N-siloxyenamines (route 1) as longer steps,
and developed the new route via [3,3]-rearrangement of N-acyloxyenamines (route 2) as shorter route.
Finally, the authors synthesised biological active (-)-CMPO, effeciently. The epimerization of acetal position was well-explained.
I think this manuscript might meet the level of "Molecules". Therefore, I recommend that this manuscript can be suitable for the publication as it stands.

Author Response

We would like to thank you for considering our manuscript and for your positive evaluation.